# A broadly generalizable stabilization strategy for sarbecovirus fusion machinery vaccines

Jimin Lee[1], Cameron Stewart [1], Alexandra Schäfer [2], Elizabeth M. Leaf[1,3], Young-Jun Park [1,4], Daniel Asarnow[1], John M. Powers [2], Catherine Treichel[1,3], Kaitlin R. Sprouse[1,4], Davide Corti [5], Ralph Baric [2], Neil P. King [1,3] & David Veesler [1,4] ✉

Evolution of SARS-CoV-2 alters the antigenicity of the immunodominant spike (S) receptor-binding domain and N-terminal domain, undermining the efficacy of vaccines and antibody therapies. To overcome this challenge, we set out to develop a vaccine focusing antibody responses on the highly conserved but metastable $S_2$ subunit, which folds as a spring-loaded fusion machinery. We describe a strategy for prefusion-stabilization and high yield recombinant production of SARS-CoV-2 $S_2$ trimers with native structure and antigenicity. We demonstrate that our design strategy is broadly generalizable to sarbecoviruses, as exemplified with the SARS-CoV-1 (clade 1a) and PRD-0038 (clade 3) $S_2$ subunits. Immunization of mice with a prefusion-stabilized SARS-CoV-2 $S_2$ trimer elicits broadly reactive sarbecovirus antibodies and neutralizing antibody titers of comparable magnitude against Wuhan-Hu-1 and the immune evasive XBB.1.5 variant. Vaccinated mice were protected from weight loss and disease upon challenge with XBB.1.5, providing proof-of-principle for fusion machinery sarbecovirus vaccines.

Several COVID-19 vaccines have been authorized worldwide to induce antibody responses targeting the SARS-CoV-2 spike (S) glycoprotein[1-3]. These vaccines enabled safe and effective protection against infection with the Wuhan-Hu-1 (Wu) isolate, which swept the globe at the beginning of the COVID-19 pandemic. However, continued viral evolution led to the emergence of SARS-CoV-2 variants with distinct antigenic properties, relative to previous isolates, eroding neutralizing antibody responses[4]. As a result, breakthrough infections have become common[5-9] although vaccinated individuals remain protected from severe disease[10-15]. Furthermore, the neutralizing activity of monoclonal antibody therapies has been compromised by these antigenic changes, resulting in the withdrawal of their regulatory authorization.

The SARS-CoV-2 S glycoprotein receptor-binding domain (RBD) is targeted by a vast diversity of antibodies and RBD-directed antibodies account for most of the plasma-neutralizing activity against infection/vaccine-matched and mismatched viruses[16-18]. Conversely, the S N-terminal domain (NTD) is mostly targeted by variant-specific neutralizing antibodies[6,19,20]. The SARS-CoV-2 $S_2$ subunit (fusion machinery) is much more conserved (Fig. 1a, Table 1) than the $S_1$ subunit (comprising the RBD and NTD), and harbors several antigenic sites targeted by broadly reactive monoclonal antibodies, including the stem helix[21-23], the fusion peptide[24-26] and the trimer apex[27]. Although some of these antibodies have neutralizing activity against a wide range of variants and distantly related coronaviruses, and protect small animals against viral challenge, their potency is limited compared to best-in-class RBD-directed antibodies[28-32]. Furthermore, fusion machinery-directed antibodies are rare in the plasma and memory B cells of previously infected and/or vaccinated subjects and

[1]Department of Biochemistry, University of Washington, Seattle, Washington, USA. [2]Department of Epidemiology, University of North Carolina, Chapel Hill, NC, USA. [3]Institute for Protein Design, University of Washington, Seattle, WA, USA. [4]Howard Hughes Medical Institute, Seattle, WA, USA. [5]Humabs Biomed SA, a subsidiary of Vir Biotechnology, Bellinzona, Switzerland. ✉e-mail: dveesler@uw.edu

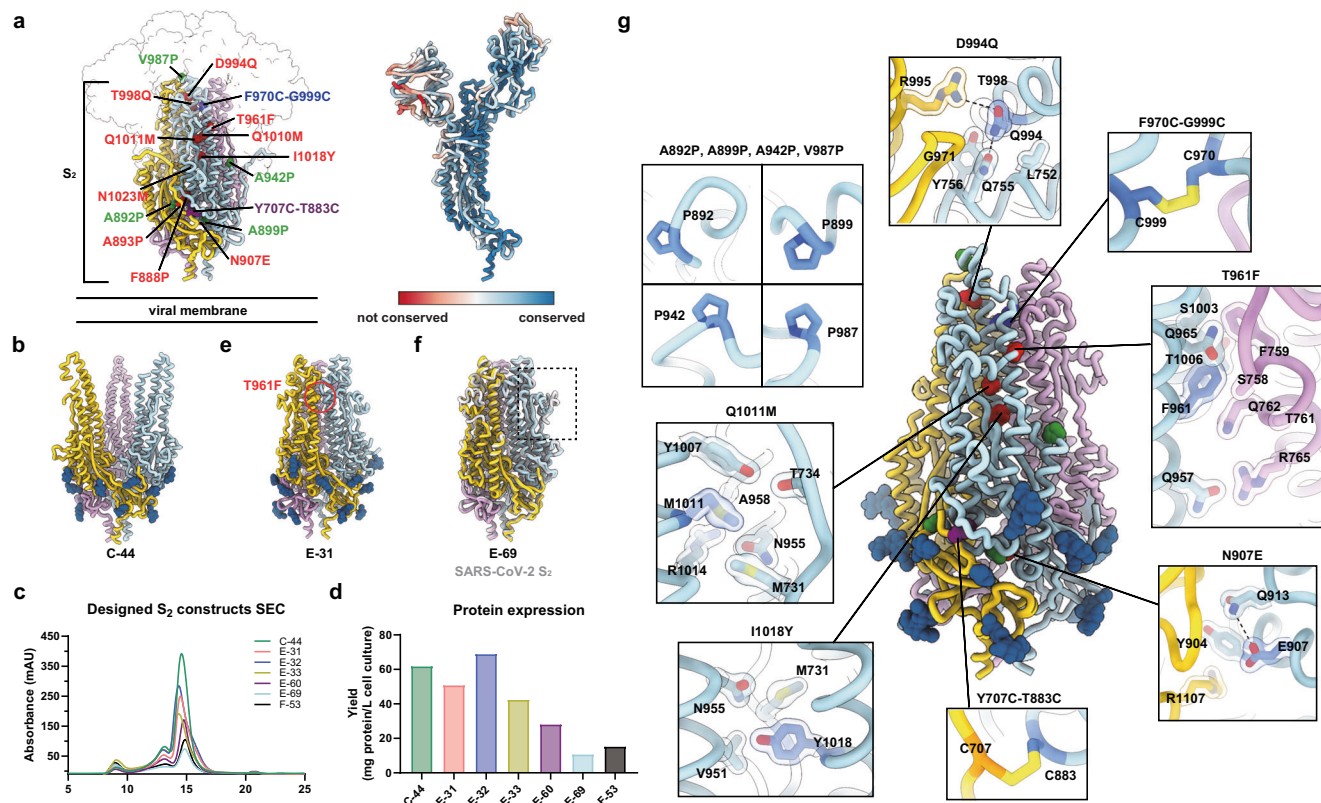

**Fig. 1 | Design of prefusion-stabilized SARS-CoV-2 fusion machinery (S$_2$ subunit) vaccines. a** (Left) Ribbon diagram of prefusion SARS-CoV-2 S highlighting all positions that were mutated to attempt to stabilize the metastable fusion machinery (S$_2$ subunit) in the prefusion conformation. Mutations are shown in blue (intra-protomer disulfide bond), purple (VFLIP inter-protomer disulfide bond[37]), green (subset of proline mutations selected from HexaPro[36]), and red (ten mutations selected based on expression/fusion score of a deep-mutational scan[42]). The S$_1$ subunit is shown as a transparent surface and glycans are omitted for clarity (PDB 6VXX). (Right) SARS-CoV-2 S (PDB 6VXX) colored by sequence conservation across multiple sarbecoviruses. **b** Ribbon diagram of the C-44 cryoEM structure previously determined with splayed open apex[17] (PDB 8DYA). **c** Size-exclusion chromatograms (SEC) of the designed S$_2$ constructs. **d** Purification yields of the designed S$_2$ constructs after size-exclusion chromatography. **e** Ribbon diagram of the E-31 cryoEM structure. The position of the T961F mutation is circled red in one protomer. **f** Superimposition of the S$_2$ subunits from the E-69 cryoEM structure and prefusion SARS-CoV-2 S[41] (gray, PDB 6XR8, residues 705-1146). The box denotes a region of local structural deviation downstream of the fusion peptide (residues 833-855). Glycans are omitted for clarity. **g** Ribbon diagram of the E-69 cryoEM structure. Insets: zoomed-in views of the mutations introduced with mutated residues rendered with darker blue/orange shades with semi-transparent surface representation of select side chains. Selected polar interactions are shown as dashed lines. The three protomers of each trimer are colored blue, pink and gold throughout the figure.

have limited contribution to neutralization mediated by polyclonal antibodies[17,21,32]. Therefore, vaccines enabling to overcome this challenge through elicitation of high titers of neutralizing antibodies targeting the conserved S$_2$ subunit bear the promise to limit the need for vaccine updates.

Here, we set out to design a prefusion-stabilized SARS-CoV-2 S$_2$ subunit vaccine enabling robust elicitation of antibodies targeting conserved fusion machinery antigenic sites to limit the impact of viral evolution on immune responses. Coronavirus S glycoproteins fold as spring-loaded trimers transitioning from the prefusion to the postfusion state, upon receptor engagement and proteolytic activation to promote membrane fusion and viral entry[33–35]. This metastability, however, constitutes a challenge for producing the S$_2$ subunit (fusion machinery) in the prefusion state in the absence of the S$_1$ subunit. Prior work described prefusion-stabilizing mutations improving the biophysical properties of the S ectodomain trimer[36–41]. Furthermore, we previously designed a fusion machinery (S$_2$ subunit) antigen[17] stabilized through the introduction of 4 out of 6 HexaPro mutations (A892P, A899P, A942P and V987P)[36], the VFLIP inter-protomer disulfide (Y707C-T883C)[37], an intra-protomer disulfide (F970C-G999C) and a C-terminus foldon trimerization domain (Fig. 1a). The resulting construct (designated C-44) harbored protomers with a prefusion tertiary structure but a quaternary structure in which the viral

membrane distal region (apex) was splayed open compared to prefusion SARS-CoV-2 S[17] (Fig. 1b). To design a prefusion S$_2$ subunit trimer with native quaternary structure, we identify a set of mutations allowing high-yield recombinant production of fusion machinery trimers with native prefusion architecture and antigenicity that remained stable in various storage conditions, as revealed through structural and antigenic studies. We further show that the prefusion-stabilization strategy designed is broadly generalizable to sarbecovirus S$_2$ subunits and successfully ported the identified mutations to the SARS-CoV-1 and PRD-0038 fusion machinery. Immunization of mice with a designed SARS-CoV-2 fusion machinery trimer vaccine elicits broadly reactive sarbecovirus antibodies and neutralizing antibody titers of comparable magnitude against the Wu and the immune evasive XBB.1.5 variant. Vaccinated mice were protected from weight loss and disease upon challenge with SARS-CoV-2 XBB.1.5 motivating further development of this vaccine.

## Results

### Design of prefusion-stabilized SARS-CoV-2 S$_2$ subunit vaccines

To design a prefusion S$_2$ subunit trimer with a native quaternary structure, we selected mutations from a deep-mutational scanning dataset of cell-surface expressed S using a library spanning residues 883 to 1034[42]. Mutations were ranked according to their expression/

**Table 1 | List of selected sarbecovirus sequences used for conservation analysis**

| Virus name | Accession Code |
|---|---|
| SARS-CoV-2 | MN908947 |
| RshSTT182 | EPI_ISL_852604 |
| GD-Pangolin | MT121216.1 |
| RaTG13 | MN996532 |
| GX-Pangolin | EPI_ISL_410542 |
| SARS-CoV-1 GD01 | AY278489 |
| SARS-CoV-1 Urbani | AY278741 |
| SARS-CoV-1 BJ01 | AY278488.2 |
| SARS-CoV-1 SZ1 | AY304489 |
| WIV1 | KF367457 |
| RsSHC014 | KC881005 |
| HuB2013 | KJ473814 |
| Rs4081 | KY417143 |
| HKU3-1 | DQ022305 |
| RacCS203 | MW251308 |
| Rf4092 | KY417145 |
| Khosta-1 | MZ190137 |
| PRD-0038 | QTJ30153.1 |
| BtKY72 | KY352407 |
| RhGB01 | MW719567 |
| Khosta-2 | MZ190138 |
| RsYN04 | EPI_ISL_1699444 |

**Table 2 | Mutations tested in the SARS-CoV-2 S$_2$ prefusion designed constructs**

| Design | Residue range | HexaPro | VFLIP inter-protomer disulfide bonds | Intra-protomer disulfide bonds |
|---|---|---|---|---|
| C-44 | 686-1208 | A892P, A899P, A942P, V987P | Y707C-T883C | F970C-G999C |

| Design | Residue range | Additional mutations to the C-44 background |
|---|---|---|
| E-31 | 686-1208 | T961F |
| E-32 | 686-1208 | D994E |
| E-33 | 686-1208 | Q1005R |
| E-60 | 686-1208 | F888P, A893P, N907E, T961F, D994Q, T998Q, Q1010M, Q1011M, I1018Y, N1023M |
| E-69 | 686-1208 | N907E, T961F, D994Q, Q1011M, I1018Y |
| F-53 | 701-1208 | N907E, T961F, D994Q, Q1011M, I1018Y |
| SARS1 S$_2$ | 683-1190 | N889E, T943F, D976Q, Q993M, I1000Y |
| PRD-0038 S$_2$ | 684-1191 | N890E, V940Q, T944F, D977Q, Q994M, I1001Y |

**Table 3 | CryoEM data collection and refinement statistics**

| | E-31 | E-60 | E-69 |
|---|---|---|---|
| **Data collection and processing** | EMD-43435 PDB 8VQ9 | EMD-43436 PDB 8VQA | EMD-43437 PDB 8VQB |
| **Magnification** | 105,000 | 105,000 | 105,000 |
| **Voltage (kV)** | 300 | 300 | 300 |
| **Electron exposure (e⁻/Å²)** ($e^-/Å^2$) | 63 | 63 | 63 |
| **Defocus range (μm)** | −0.2 - −7.0 | −0.2 - −3.0 | −0.1 - −3.27 |
| **Pixel size (Å)** | 0.843 | 0.843 | 0.843 |
| **Symmetry imposed** | C3 | C3 | C3 |
| **Final particle images (no.)** | 671,707 | 144,044 | 319,001 |
| **Map resolution (Å)** | 2.7 | 3.5 | 3.0 |
| **FSC threshold** | 0.143 | 0.143 | 0.143 |
| **Map sharpening B factor ($Å^2$)** | −119.4 | −143.9 | −127.6 |
| **Validation** | | | |
| **MolProbity score** | 0.94 | 0.96 | 0.89 |
| **Clashscore** | 0.94 | 0.78 | 0.65 |
| **Poor rotamers (%)** | 0.00 | 0.77 | 0.71 |
| **Ramachandran plot** | | | |
| **Favored (%)** | 97.12 | 96.68 | 96.95 |
| **Allowed (%)** | 2.88 | 2.61 | 3.05 |
| **Disallowed (%)** | 0.00 | 0.71 | 0.00 |

protein per liter of Expi293 cells (Fig. 1c, d). Single-particle electron microscopy (EM) characterization of each negatively stained glycoprotein revealed that the design harboring the T961F mutation (E-31) formed prefusion closed S$_2$ trimers whereas the other two constructs, harboring either D994E (E-32) or Q1005R (E-33), adopted several conformations, including one with a splayed-open apex (Supplementary Fig. 1) similar to C-44[17]. Furthermore, we observed a higher tendency to aggregate for E-32 and E-33 by SEC and EM (Fig. 1c and Supplementary Fig. 1). To unveil the E-31 architecture and understand the effect of the T961F mutation, we determined a cryoEM structure of this antigen at 2.7 Å resolution (Fig. 1e, Supplementary Fig. 2 and Table 3). Our structure shows that E-31 folds with virtually identical tertiary and quaternary structures to the S$_2$ subunit from the prefusion S ectodomain trimer (PDB 6VXX[41]) with which the protomers can be superimposed with r.m.s.d values of 0.7 Å. The engineered T961F substitution, which maps to the distal half of the fusion machinery apex, reinforces interactions between HR1 and the central helix of one protomer and the upstream helix of a neighboring protomer (cavity filling), likely explaining the compact, closed trimer conformation observed (Supplementary Fig. 3a). Although we could detect a small fraction of E-31 trimers with an open apex conformation in the cryoEM dataset, our findings demonstrate that the T961F mutation alone is sufficient to close the fusion machinery apex in a prefusion conformation (Fig. 1e, Supplementary Fig. 2).

To further improve the conformational homogeneity of our vaccine candidate, we designed two additional constructs comprising either 4 or 9 additional residue substitutions added to the E-31 trimer (Table 2). Both constructs were recombinantly produced to characterize their expression, stability, and structural properties (Fig. 1c, d). The E-60 construct harbors all 9 additional mutations we selected to add to the E-31 scaffold. Although recombinant production of this protein construct led to high expression levels, its cryoEM structure at 3.5 Å resolution (Supplementary Fig. 4, Table 3) revealed that the introduction of several proline mutations in the region comprising residues 879–897 distorted an α-helix and following loop relative to its native structure (Supplementary Fig. 3b). Ruling out conformation distorting-mutations and a few additional mutations based on visual inspection, we designed a new construct which we named E-69 and

fusion score ratio, to favor prefusion-stabilizing amino acid substitutions and visually inspected to lead to a final selection of ten mutations (Fig. 1a, Table 2). We first evaluated the effect of three individual mutations introduced in the C-44 background, namely T961F, D994E, and Q1005R, which were among the highest ranking based on their expression/fusion score ratio. We recombinantly produced each of the three constructs (designated E-31, E-32 and E-33 for the T961F, D994E and Q1005R mutations, respectively) in human cells and characterized their monodispersity by size-exclusion chromatography (SEC). Residue 961 is part of the heptad repeat 1 (HR1) motif whereas residues 994 and 1005 are located in the central helix (CH)[33,34,41]. All three constructs eluted mainly as monodisperse species, yielding 40–60 mg of purified

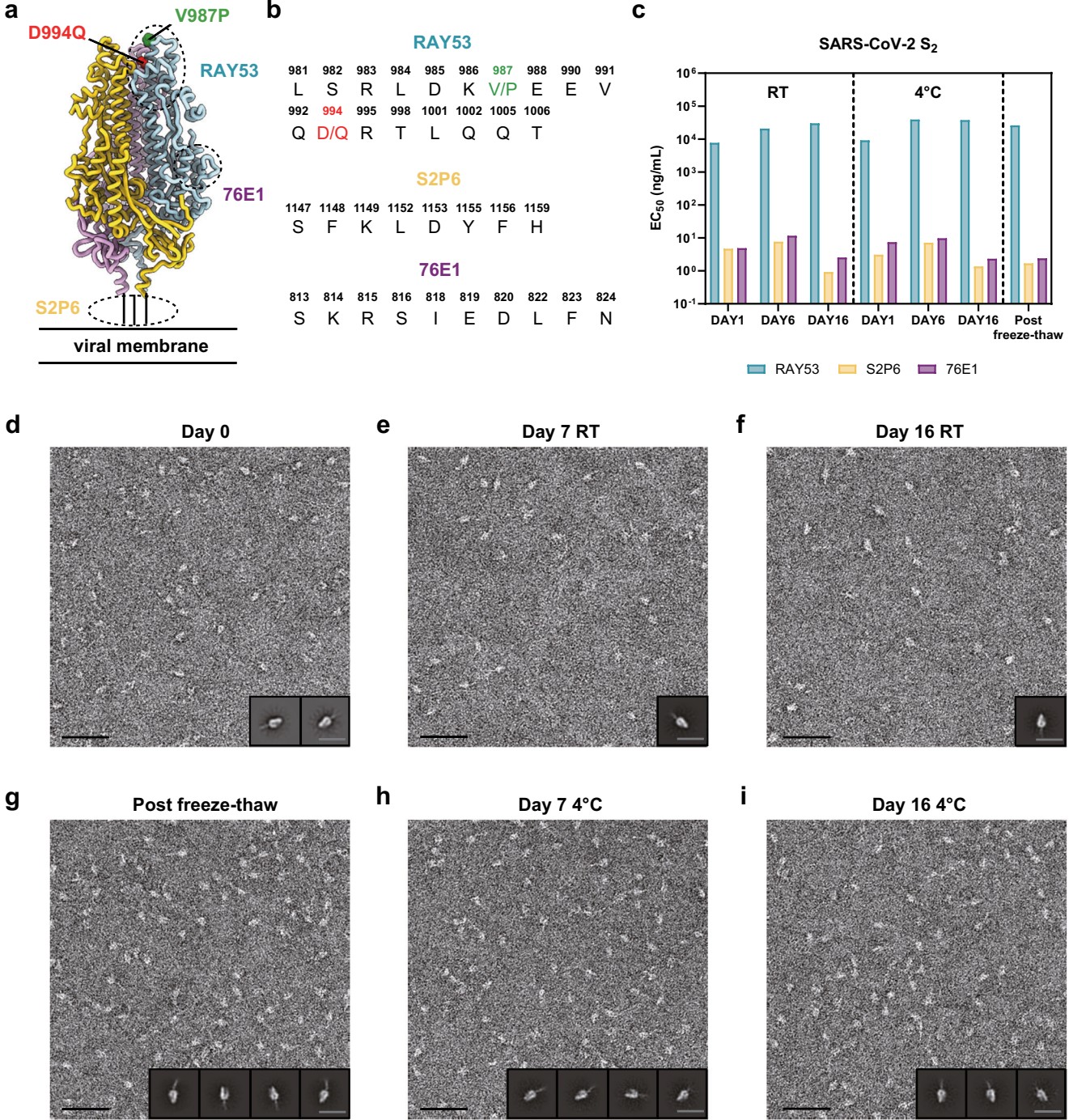

**Fig. 2 | A stable prefusion-stabilized SARS-CoV-2 fusion machinery (S₂ subunit) vaccine candidate. a** Ribbon diagram of the E-69 prefusion-stabilized SARS-CoV-2 fusion machinery antigen highlighting the regions containing the epitopes recognized by S₂ subunit-targeting monoclonal antibodies using dashed lines. **b** Amino acid sequences of the RAY53, S2P6 and 76E1 epitopes. The V987P and D994Q E-69 mutations are respectively shown in green and red in panels A-B as they are located within the RAY53 epitope that has been previously reported[27]. **c** Evaluation of retention of antigenicity for the E-69 antigen in various storage conditions using binding of the S2P6, 76E1 and RAY53 monoclonal antibodies analyzed by ELISA. **d**–**i** Evaluation of retention of the native prefusion conformation for the E-69 antigen in various storage conditions analyzed by EM of negatively stained samples. Insets: 2D class averages. The scale bars represent 50 nm (micrograph) and 200 Å (2D class averages). nsEM data was collected once per storage conditions. Representative micrographs from 42, 85, 49, 42, 22, and 24 micrographs are shown respectively. Particles picked from individual sets of micrographs were used to generate 2D class averages.

harbors the N907E, D994Q, Q1011M and I1018Y in the E-31 background. To investigate its atomic-level organization, we determined a cryoEM structure of E-69 at 3 Å resolution and observed clear cryoEM density defining side chains of most amino acid mutations introduced and confirming folding as a prefusion closed S₂ subunit trimer, without particle images corresponding to trimers with an open apex (Fig. 1f and

Supplementary Fig. 5). The overall E-69 architecture is nearly identical to that of the S₂ subunit trimer from the prefusion S ectodomain structure (PDB 6VXX[41]) (Fig. 1f). Only one region noticeably deviated from prefusion S in terms of local structure downstream of the fusion peptide (residues 833-855) (Fig. 1f and Supplementary Fig. 3c). In the intact SARS-CoV-2 S trimer, this region makes contact with the S₁

subunit C and D domains of a neighboring protomer, which are absent in our design, likely allowing adoption of an altered local conformation, as is the case in all our constructs (Supplementary Fig. 3c). The N907E substitution places the newly introduced side chain carboxylate close to the Q913 side chain amide with which it interacts electrostatically (Fig. 1g). The D994Q side chain amide forms an intraprotomer hydrogen bond with the Q755 side chain amide and an interprotomer electrostatic interaction with the R995 guanidinium, thereby strengthening interactions at the distal part of the apex (Fig. 1g). The Q1011M mutation is likely stabilizing through the reinforcement of local hydrophobic packing with the nearby M731 and Y1007 residues (Fig. 1g). The I1018Y mutation fills a cavity within each protomer through local interactions involving the CH, HR1, and upstream helix region (Fig. 1g).

As our E-69 structure did not resolve the flexible N-terminal region, which connects the $S_1$ and $S_2$ subunits in the context of the S trimer, we designed a new construct lacking 15 residues at the N-terminus of E-69 and named it F-53 (Fig. 1c, d, Table 2). F-53 expressed better than E-69 and retained identical antigenicity, and thus served as a template for subsequent $S_2$ antigen design (Supplementary Fig. 6).

## A stable prefusion-stabilized SARS-CoV-2 fusion machinery antigen

Storage and shipping conditions are important considerations for vaccine design and manufacturing as they can impact the cost of goods, ease of distribution and in turn global access. To evaluate the stability of the prefusion-stabilized SARS-CoV-2 E-69 design, we investigated the retention of antigenicity after storage in various conditions by ELISA using a panel of monoclonal antibodies targeting the $S_2$ subunit (Fig. 2a, b). The stem helix-targeting S2P6 antibody[21], the fusion peptide-directed 76E1 antibody[25], and the fusion machinery apex-recognizing RAY53[27] bound to E-69 (Fig. 2a–c), as was the case with SARS-CoV-2 S. E-69 retained unaltered antigenicity for at least two weeks at room temperature and at 4 °C, and could be frozen and thawed without affecting its antigenicity (Fig. 2c and Supplementary Fig. 7). To assess biophysical stability, we analyzed purified E-69 at various time points using negative staining EM. 2D class averages of E-69 showed retention of its prefusion conformation for at least two weeks both at room temperature and at 4 °C and that it could be frozen and thawed without altering its structure (Fig. 2d–i). Although we detected a minor population of trimers with an open apex upon storage at low temperatures (Fig. 2g–i), the 2D class averages suggested that the magnitude of these structural changes might be smaller than that observed with the C-44, E-32 or E-33 designed constructs (Fig. 1b, Supplementary Fig. 1c, 1d). Furthermore, we did not detect any 2D class averages corresponding to open trimers in our E-69 cryoEM dataset (Supplementary Fig. 5). These data suggest that the E-69 design is stable, highlighting the robustness of our prefusion-stabilization strategy, and endowed with optimal biophysical properties for a vaccine candidate.

## A broadly generalizable prefusion-stabilization strategy for sarbecovirus fusion machinery immunogens

To assess the broad applicability of our $S_2$ subunit prefusion-stabilization strategy, we examined the local structural environment of the residues mutated in E-69 and compared it with the corresponding regions of interest in SARS-CoV-1 $S_2$ (clade 1a) and PRD-0038 $S_2$ (clade 3)[43]. The overall architecture of the SARS-CoV-1 $S_2$ and PRD-0038 $S_2$ trimers is similar to that of SARS-CoV-2 $S_2$ with which they can be superimposed with r.m.s.d. values of 1.1 and 1.3 Å, and share 90% and 87% amino acid sequence identity, respectively. Based on the observed conservation of the local structural environment, we hypothesized that the E-69 mutations should be portable to SARS-CoV-1 $S_2$ and to PRD-0038 $S_2$ and designed the corresponding constructs

(Fig. 3a–i). We additionally truncated the N-terminal region of the prefusion-stabilized SARS-CoV-1 $S_2$ and PRD-0038 $S_2$ constructs due to the enhanced expression of SARS-CoV-2 F-53 relative to E-69 (Fig. 1c, d). Finally, we introduced the V940Q substitution to the PRD-0038 $S_2$ construct, which is the position equivalent to SARS-CoV-2 Q957 and SARS-CoV-1 Q939, to allow electrostatic interaction with R748 (similar to that observed with SARS-CoV-2 R765 and SARS-CoV-1 R747 from a neighboring protomer) (Fig. 3d).

Recombinant production and purification of the designed SARS-CoV-1 and PRD-0038 $S_2$ subunit constructs led to even greater yields than that of SARS-CoV-2 F-53, reaching 50 mg of purified SARS-CoV-1 $S_2$ trimer per liter of Expi293 cells (Fig. 3j, k). These two trimers were stable in a variety of storage conditions and retained unaltered reactivity with fusion machinery-directed monoclonal antibodies for at least two weeks at room temperature and at 4 °C and could be frozen and thawed without affecting their antigenicity (Fig. 3l, m, and Supplementary Fig. 8). Although the RAY53 and S2P6 antibodies each cross-reacted with comparable efficiency to the SARS-CoV-1 $S_2$ trimer and to the PRD-0038 $S_2$ trimer, 76E1 bound efficiently to SARS-CoV-1 $S_2$ but much more weakly to PRD-0038 $S_2$, possibly as a result of the $F823_{SARS-CoV-2}Y806_{PRD-0038}$ epitope mutation[43] (Fig. 3l, m). Single particle EM analysis of negatively stained SARS-CoV-1 $S_2$ and PRD-0038 $S_2$ showed that they adopt the designed closed prefusion architecture (Fig. 3n, o). Collectively, these data indicate that our $S_2$ prefusion-stabilization strategy is broadly applicable across different sarbecovirus clades and promotes retention of native structure and antigenicity over time under various storage conditions.

## A prefusion-stabilized SARS-CoV-2 fusion machinery vaccine elicits broadly reactive antibody responses

To evaluate the immunogenicity of our lead prefusion SARS-CoV-2 $S_2$ designed vaccine candidate, we immunized twelve BALB/c mice with four 5 µg doses of E-69 on weeks 0, 3, 10 and 17 and twelve BALB/c mice with two 1 µg doses of prefusion SARS-CoV-2 2P S on weeks 0 and 3 followed by two 5 µg doses of E-69 on weeks 10 and 17. The latter group aims to recapitulate the pre-existing immunity found in humans due to prior exposures through vaccination and/or infection. For benchmarking, we also immunized twelve BALB/c mice with four 1 µg doses of prefusion SARS-CoV-2 2P S on weeks 0, 3, 10, and 17 (Fig. 4a). All immunogens were adjuvanted with Addavax using a 1:1 (v/v) ratio.

Binding antibody titers were analyzed by ELISA using sera obtained two weeks post dose 4 (week 19) against SARS-CoV-2 Wu HexaPro S, SARS-CoV-2 XBB.1.5 HexaPro S, and SARS-CoV-1 HexaPro S (Fig. 4b–d and Supplementary Fig. 9). We observed similar prefusion SARS-CoV-2 Wu S and XBB.1.5 S antibody binding titers upon immunization with E-69 (GMT: 5.2/4.9), SARS-CoV-2 2P S (GMT: 5.2/5.0) or SARS-CoV-2 2P S followed by E-69 (GMT: 5.4/5.0). However, we observed slightly higher SARS-CoV-1 S binding titers with the E-69 (GMT: 5.2) or SARS-CoV-2 2P S followed by E-69 (GMT: 5.0) vaccination regimens, as compared to SARS-CoV-2 2P S immunization (GMT: 4.5). These results suggest that E-69 vaccination elicits comparable serum antibody binding titers after four doses to the widely used prefusion 2P S trimer against SARS-CoV-2 variants but slightly higher titers against SARS-CoV-1 in these experimental conditions. ELISA analysis of vaccine-elicited serum antibodies five weeks post dose 4 (week 22) targeting SARS-CoV-2 E-69 showed that immunization with E-69 or with E-69 followed by SARS-CoV-2 2P S (GMTs: 4.9/4.8) led to slightly higher IgG binding titers than SARS-CoV-2 2P S immunization (GMT: 4.4), providing proof-of-principle for specific elicitation of fusion machinery-directed antibodies (Supplementary Fig. 10). Moreover, E-69 vaccination elicited a broad spectrum of IgG subclasses, including IgG1, IgG2a and IgG2b, that was comparably balanced to that induced by SARS-CoV-2 2P S vaccination (Supplementary Fig. 10).

We subsequently evaluated vaccine-elicited neutralizing activity using sera obtained two weeks post dose 4 (week 19) and vesicular

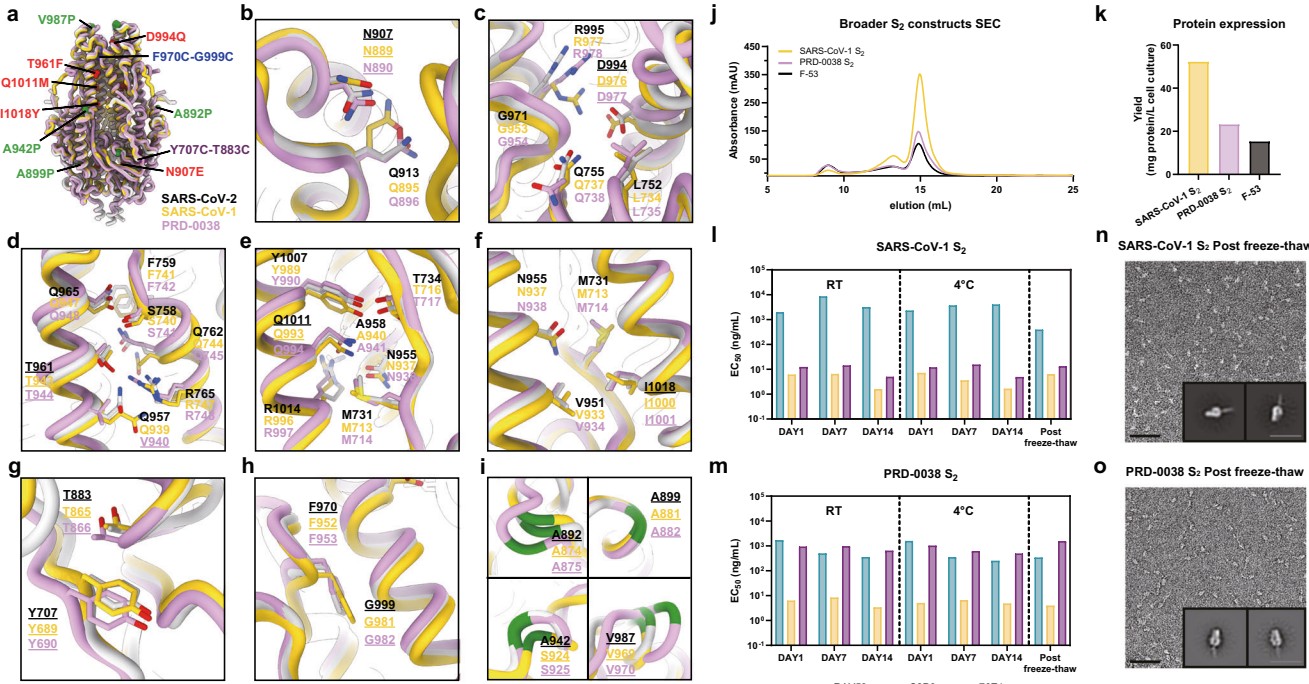

**Fig. 3 | A broadly generalizable prefusion-stabilization strategy for sarbecovirus fusion machinery (S₂ subunit) antigens. a** Ribbon diagrams of superimposed S₂ subunits of the prefusion SARS-CoV-2 S (PDB 6VXX[41]), SARS-CoV-1 S (PDB 5X58[82]) and PRD-0038 S (PDB 8U29[43]) structures. Prefusion-stabilizing mutations are shown in blue (intra-protomer disulfide bond), purple (VFLIP inter-protomer disulfide bond[37]), green (subset of proline mutations selected from HexaPro[36]), and red (mutations ported from E-69). **b–i** Zoomed-in views of superimposed S₂ subunits of the prefusion SARS-CoV-2, SARS-CoV-1 and PRD-0038 S structures highlighting the local structural conservation of residues mutated in SARS-CoV-2 the E-69/F-53 constructs (underlined). SARS-CoV-2, SARS-CoV-1, and PRD-0038 S are respectively shown in light gray, gold, and pink in panels (**a–i**). **j** Size-exclusion chromatograms (SEC) of the designed SARS-CoV-1 and PRD-0038 S₂ constructs, as compared to SARS-CoV-2 F-53. **k** Purification yields of the designed SARS-CoV-1 and PRD-0038 S₂ constructs. The yield for the best SARS-CoV-2 S₂ construct (F-53) is included for comparison. **l, m** Evaluation of retention of antigenicity for the SARS-CoV-1 (**l**) and PRD-0038 (**m**) S₂ antigens in various storage conditions using binding of the S2P6, 76E1 and RAY53 monoclonal antibodies analyzed by ELISA. **n, o** Evaluation of retention of the native prefusion conformation of the negatively stained SARS-CoV-1 (**n**) and PRD-0038 (**o**) S₂ trimers after freeze/thawing. Insets: 2D class averages showing compact prefusion S₂ trimers. The scale bar represents 50 nm (micrographs) and 200 Å (2D class averages). nsEM data was collected once per storage conditions. Representative micrographs from 82 and 98 micrographs are shown respectively. Particles picked from individual sets of micrographs were used to generate 2D class averages.

stomatitis virus (VSV) particles pseudotyped with SARS-CoV-2 Wu/G614, XBB.1.5 S or SARS-CoV-1 S (Fig. 4e−g and Supplementary Fig. 11). SARS-CoV-2 2P S or SARS-CoV-2 2P S followed by E-69 vaccination elicited potent neutralizing antibodies against the vaccine-matched SARS-CoV-2 Wu/G614 whereas E-69 elicited modest neutralizing activity. SARS-CoV-1 S VSV neutralization was highest upon vaccination with SARS-CoV-2 2P S and comparably lower for the groups vaccinated with E-69 or SARS-CoV-2 2P S followed by E-69. Neutralization of XBB.1.5 VSV S, however, was marginally higher for mice vaccinated with E-69 or with SARS-CoV-2 2P S followed by E-69 as compared to vaccination with SARS-CoV-2 2P S.

**A prefusion-stabilized SARS-CoV-2 fusion machinery vaccine protects against the SARS-CoV-2 XBB.1.5 variant**
To study the in vivo protective efficacy of the designed fusion machinery SARS-CoV-2 vaccine against an immune evasive SARS-CoV-2 variant, we intranasally inoculated each animal in the three aforementioned vaccinated groups of mice with 10⁵ PFU of XBB.1.5 MA10[44] (Fig. 4a). We also challenged 10 unvaccinated mice as a control group. Weight loss was followed for 4 days post infection (Fig. 5a) whereas replicating viral titers in the nasal turbinates and lung as well as lung pathology were assessed at 2 and 4 days post challenge (Fig. 5b–d). Although none of the immunogens evaluated protected from infection, likely due to the systemic delivery route[13,45–47], all vaccinated mice had markedly reduced weight loss throughout the duration of the experiment, as compared to unvaccinated mice (Fig. 5a, b). Furthermore, lung viral load and lung pathology were comparable for all three

vaccinated groups and greatly improved compared to unvaccinated animals (Fig. 5c, d). These data indicate that vaccination with our lead prefusion-stabilized SARS-CoV-2 S₂ subunit (fusion machinery) immunogen elicited protection against disease induced by the highly immune evasive SARS-CoV-2 XBB.1.5 variant in this stringent challenge model.

## Discussion
The emergence of immune evasive SARS-CoV-2 variants erodes the effectiveness of COVID-19 vaccines, which led to the roll out of two updated boosters in 2022[48,49] and 2023[50]. For the foreseeable future, it is likely that COVID-19 vaccines will require yearly reformulation based on the anticipated prevalence of circulating variants, similarly to influenza virus vaccines. Next-generation vaccines that are more resilient to viral evolution and antigenic changes bear the promise of reducing the need for or the frequency of vaccine updates, which would also help with large-scale adoption by the public. The sarbecovirus S₂ subunit prefusion-stabilization strategy presented here represents a key step in this direction due to the much higher conservation of the S₂ subunit relative to the S₁ subunit among SARS-CoV-2 variants and other sarbecoviruses[14,41,51]. Given the limited potency of known fusion machinery-directed monoclonal antibodies[21,22,24,25,27], relative to S₁-targeting antibodies, the plasma neutralizing activity induced by SARS-CoV-2 S₂ vaccination was weaker than that of prefusion 2P S although its breadth was much less affected by antigenic changes. In vivo evaluation of vaccine efficacy suggests that SARS-CoV-2 S₂ protected mice comparably to prefusion 2P S, based on weight

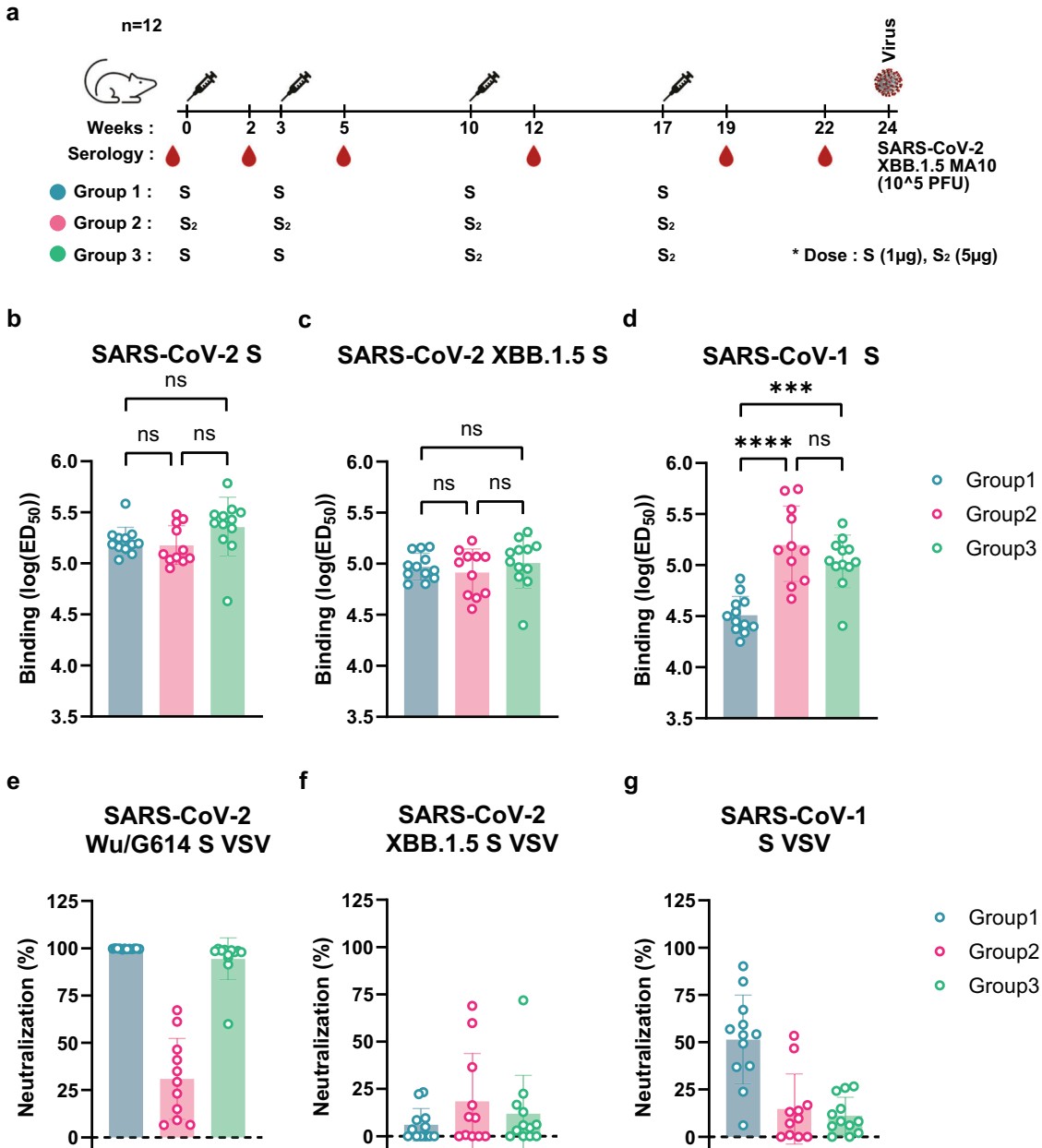

**Fig. 4 | A prefusion-stabilized SARS-CoV-2 fusion machinery (S₂ subunit) vaccine elicits broadly reactive antibody responses. a** Vaccination schedule and study design. **b**–**d** Analysis of antibody binding titers against SARS-CoV-2 Hexapro S (**b**), XBB.1.5 Hexapro S (**c**), and SARS-CoV-1 Hexapro S (**d**) analyzed by ELISAs using sera obtained two weeks post dose 4. Geometric mean and geometric standard deviation (SD) are shown as bars. Each data point represents the mean of two biological replicates each comprising two technical replicates. Comparisons between multiple groups for (**b**–**d**) were made by ordinary one-way analysis of variance (ANOVA) followed by Tukey's multiple comparisons test. ns:$P > 0.05$,

$*:P < = 0.05$, $**:P < = 0.01$, $***:P < = 0.001$, $****:P < = 0.0001$. $P = 2.645e-006$ and $P = 0.0001505$ for the comparisons made between groups1&2 and groups 1&3, respectively. **e**–**g** Analysis of neutralizing antibody titers expressed as percentage of neutralization using a 1/33 dilution of sera obtained two weeks post dose 4 against SARS-CoV-2 Wu/G614 (**e**), XBB.1.5 (**f**), and SARS-CoV-1 (**g**) S VSV pseudoviruses. Each data point represents the mean values from three biological replicates. Means and SDs for each group are shown as bars ($n = 12$ for groups1&3 and $n = 11$ for group2). Representative dose-response curves are shown in Supplementary Fig. 11.

loss and viral replication in the upper and lower airways, upon challenge with the immune evasive SARS-CoV-2 XBB.1.5 variant in these experimental conditions.

We note that SARS-CoV-2 S₂ subunit vaccination protected mice that did and those that did not have detectable serum neutralizing activity against SARS-CoV-2 XBB.1.5, as observed for stabilized MERS-CoV stems upon MERS-CoV challenge[52]. Although the immunological mechanisms underlying the observed protection remain to be defined, we postulate that weakly or non-neutralizing antibodies participated in protection through Fc-mediated effector functions, as described for

the S2P6 stem-helix antibody[21] and for S-elicited fusion machinery-directed polyclonal antibodies upon mismatched sarbecovirus challenge[53]. Future studies will decipher the contribution of these distinct branches of the immune response to the protection observed.

The broadly generalizable prefusion-stabilization strategy described here provides a robust platform for elicitation of fusion machinery-directed antibody responses and the designed antigens will enable studies of such immune responses. Future engineering efforts may further improve the immunogenicity of this vaccine candidate through (i) multivalent display at the surface of a nanoparticle, as

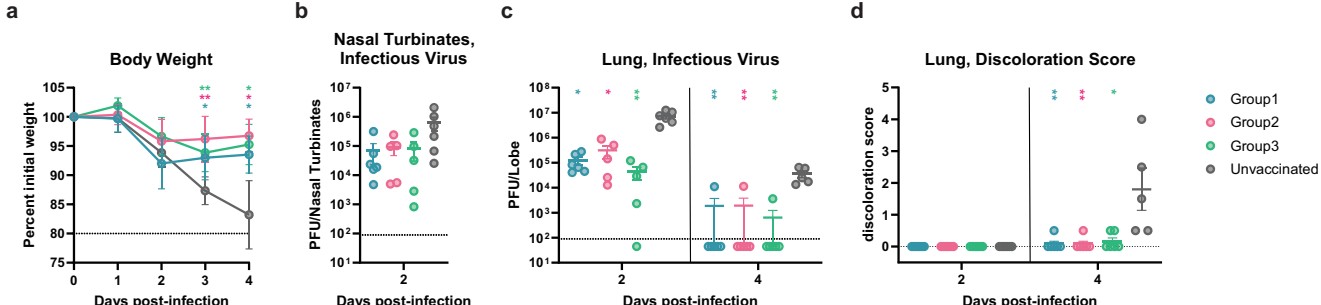

**Fig. 5 | A prefusion-stabilized SARS-CoV-2 fusion machinery (S₂ subunit) vaccine protects mice against SARS-CoV-2 XBB.1.5-induced disease. a** Weight loss followed 4 days post viral challenge with SARS-CoV-2 XBB.1.5 MA10. Control group corresponds to unvaccinated mice. Mean values and SD are shown ($n = 12$ for days 0-2 for groups1&unvaccinated and $n = 11$ for days 0-2 for groups 2&3, and $n = 6$ for days 3-4 for all groups except $n = 5$ for the unvaccinated group). Comparison of percent weight loss with the control group and each of the vaccinated groups was done using two-way analysis of variance (ANOVA) followed by Sidak's multiple comparison test. ns:$P > 0.05$, *:$P <= 0.05$, **:$P < 0.01$, ***:$P < 0.001$, ****:$P < 0.0001$. For day 3, $P = 0.0411$, $P = 0.0038$, and $P = 0.0093$ for the comparisons made between the unvaccinated and groups 1&2&3, respectively. For day 4, $P = 0.0379$, $P = 0.0120$, and $P = 0.0188$ for the comparisons made between the unvaccinated and groups 1&2&3, respectively. **b, c** Quantification of replicating viral titers in the nasal turbinates (**b**) and lungs (**c**) of challenged animals at 2 and 4 days post infection, respectively. Mean values and standard error of measurement (SEM) are shown ((dpi2);$n = 6$ for groups1&unvaccinated and $n = 5$ for groups 2&3, (dpi4);$n = 6$ for groups1&2&3 and $n = 5$ for the unvaccinated group). For the infectious virus titer in the lung 2 days post-infection, $P = 0.0262$, $P = 0.0471$, and $P = 0.0012$ for the comparisons made between the unvaccinated and groups 1&2&3, respectively. For 4 days post-infection, $P = 0.0030$, $P = 0.0035$, and $P = 0.025$ for the comparisons made between the unvaccinated and groups 1&2&3, respectively. **d** Lung discoloration score at 2 and 4 days post infection. Mean values and SEM are shown ((dpi2);n = 6 for groups1&unvaccinated and $n = 5$ for groups 2&3, (dpi4);$n = 6$ for groups1&2&3 and $n = 5$ for the unvaccinated group). For 4 days post-infection, $P = 0.0061$, $P = 0.0061$, and $P = 0.0264$ for the comparisons made between the unvaccinated and groups 1&2&3, respectively. Statistical significance compared with the control group and each of the vaccinated groups was reported using Kruskal-Wallis test followed by Dunn's multiple comparisons test (**b**–**d**). ns: $P > 0.05$, *:$P <= 0.05$, **:$P <= 0.01$, ***:$P < 0.001$, ****:$P < 0.0001$. The experiment has not been replicated (**a**–**d**).

exemplified for the SKYcovione SARS-CoV-2 RBD vaccine[54–56]; (ii) mRNA-launching of a membrane-anchored SARS-CoV-2 S₂ subunit vaccine, a vaccine platform which yielded multiple efficacious SARS-CoV-2 S 2P vaccines[1]; or (iii) a recently proposed targeted deglycosylation approach of the SARS-CoV-2 S₂ subunit to enhance neutralizing antibody titers[57].

## Methods

### Ethical Statement

Animal studies were carried out in accordance with the Department of Comparative Medicine at the University of Washington, Seattle, accredited by the Association for Assessment and Accreditation of Laboratory Animal Care (AAALAC). Animal experiments were conducted in accordance with the University of Washington's Institutional Animal Care and Use Committee (IACUC: 4470-01).

### Cell lines

Cell lines used in this study were obtained from HEK293T (ATCC, CRL-11268), Expi293F (Thermo Fisher Scientific, A145277) and VeroE6-TMPRSS2 (JCRB1819). Cells were cultured in 10% FBS, 1% penicillin-streptomycin, 2% Geneticin (applicable for Vero cells, only) DMEM at 37°C, 5% $CO_2$. None of the cell lines were authenticated or tested for mycoplasma contamination.

### Production of recombinant S₂ antigen proteins

Each S₂ construct was produced in Expi293F cells (ThermoFisher Scientific) and cultured at 37 °C in a humidified 8% $CO_2$ incubator with constant rotation at 130 RPM using Expi293 Expression Medium (ThermoFisher Scientific). DNA transfections were conducted using the ExpiFectamine 293 Transfection Kit (ThermoFisher Scientific) protocols and materials and cultivated for five days before harvest. Cell culture supernatants were clarified by centrifugation and proteins were harvested using HisTrap™ High Performance Ni Sepharose columns (Cytiva). Proteins were washed using 10–15 CVs of buffer containing 25 mM Tris, 150 mM NaCl, 20 mM Imidazole pH 8.0 followed by elution with 10–15 CVs of buffer containing 25 mM Tris, 150 mM NaCl, 300 mM Imidazole pH 8.0. Eluates were buffer exchanged and

concentrated into 20 mM Tris, 150 mM NaCl, pH 8.0 using Amicon Ultra-15 Centrifugal Filter Unit (10 kDa) (Millipore). Gel filtration was performed to remove unfolded or aggregated protein thus samples were each run through a Superose-6 Increase 10/300 GL column (Cytiva)) equilibrated in 20 mM Tris, 150 mM NaCl, pH 8.0. Main peaks were collected and protein was snap-frozen and stored at -80 °C with some set aside for stability tests. Purified proteins for immunogenicity study were tested for endotoxin levels using Limulus Amebocyte Lysate (LAL) cartridges (Charles River PTS201F).

### Production of recombinant SARS-CoV-2 2P S, HexaPro S and SARS-CoV-1 HexaPro S

The SARS-CoV-2 2P S glycoprotein ectodomain construct was previously described[41] and comprised an abrogated S₁/S₂ cleavage site (R682S, R683G and R685G)[33,35,58,59], two consecutive proline stabilizing mutations (K986P and V987P, so called 2P[38,39]) and a C-terminal foldon trimerization domain[60]. The SARS-CoV-2 HexaPro S glycoprotein ectodomain construct comprises residues 1-1208 with the native signal peptide, the HexaPro prefusion stabilizing mutations (F817P, A892P, A899P, A942P, K986P, V987P), abrogation of the S₁/S₂ cleavage site (R682G, R683S and R685S), a C-terminal short linker (GSG) followed by a foldon, HRV 3 C site (LEVLFQGP), a short linker (GSG), an avi tag, a short linker (GSG), an 8x his tag in a pcDNA3.1(-) plasmid[36]. The SARS-CoV-1 HexaPro S glycoprotein ectodomain construct comprises residues 1-1190 (UniProt P59594-1) with the native signal peptide, the HexaPro prefusion stabilizing mutations (F799P, A874P, A881P, S924P, K968P, V969P) followed by a C-terminal short linker (GSG) followed by a foldon, HRV 3 C site (LEVLFQGP), an avi tag, a short linker (GSG), an 8x his tag in a CMVR plasmid. Expi293F cells were grown at 37°C with 8% $CO_2$ and DNA transfections were conducted with the ExpiFectamine 293 Transfection Kit (Thermo Fisher Scientific). Cell culture supernatants were harvested four days post-transfection and proteins were purified using HisTrap™ High Performance column (Cytiva). Proteins were first washed with 10–15 column volumes of a buffer containing 25 mM sodium phosphate, 300 mM NaCl, 20 mM imidazole, pH 8.0, followed by elution with 10-15 column volumes using 300 mM imidazole, pH 8.0. Eluted proteins were concentrated and

buffer exchanged into 1x TBS (20 mM Tris, 150 mM NaCl, pH 8.0) using Amicon Ultra-15 Centrifugal Filter Unit (100 kDa) (Millipore). Purified proteins were snap-frozen and stored at -80 ˚C. SARS-CoV-2 2P S proteins for immunogenicity study were tested for ensuring low endotoxin levels using Limulus Amebocyte Lysate (LAL) cartridges (Charles River PTS201F).

## Monoclonal antibody ELISAs
For monoclonal antibody ELISAs, 30 µl of the proteins at 3 µg/mL were plated onto 384-well Nunc Maxisorp plate (ThermoFisher, 464718) in 1x TBS and incubated 1 h at 37 °C followed by slap drying and blocking with 80 µL of Casein for 1 h at 37 °C. After incubation, plates were slap-dried and 1:4 serial dilutions of the corresponding mAbs starting from 0.1 mg/ml were made in 30 µl TBST, added to the plate and incubated at 37 °C for 1 h. Plates were washed 4x in TBST and 30 µl of 1:5000 Goat anti-Human IgG Fc Secondary Antibody, HRP (Thermo Fisher, A18817) were added to each well and incubated at 37 °C. After 1 h, plates were washed 4x in TBST and 30 µl of TMB (SeraCare) was added to every well for 2 min at room temperature. Reactions were quenched with the addition of 30 µl of 1 N HCl. Plates were immediately read at 450 nm on a BioTek Neo2 plate reader and data was plotted and fit in Prism 9 (GraphPad) using nonlinear regression sigmoidal, 4PL, X is the concentration to determine $EC_{50}$ values from curve fits.

## Production of VSV pseudoviruses
SARS-CoV-2 D614G S, XBB.1.5 S, and SARS-CoV-1 S VSV pseudoviruses were produced using HEK293T cells seeded on BioCoat Cell Culture Dish: poly-D-Lysine 100 mm (Corning). Cells were transfected with respective S constructs using Lipofectamine 2000 (Life Technologies) in Opti-MEM transfection medium. After 5 h of incubation at 37 °C with 5% CO2, cells were supplemented with DMEM containing 10% of FBS. On the next day, cells were infected with VSV (G*ΔG-luciferase) for 2 h, followed by five time wash with DMEM medium before addition of anti-VSV G antibody (I1-mouse hybridoma supernatant diluted 1:40, ATCC CRL-2700) and medium. After 18–24 h of incubation at 37 °C with 5% CO2, pseudoviruses were collected and cell debris was removed by centrifugation at 3000 x $g$ for 10 min. Pseudoviruses were further filtered using a 0.45 µm syringe filter and concentrated 10x prior to storage at -80 °C.

## Serological ELISAs
For serological ELISAs, 30 µL of assorted proteins (SARS-CoV-2 HexaPro S, SARS-CoV-2 XBB.1.5 HexaPro S (AcroBiosystems, SPN-C524i) and SARS-CoV-1 HexaPro S) at 3 µg/mL were placed into 384-well Nunc Maxisorp plates (ThermoFisher, 464718) in 1x TBS and incubated for 1 h at 37 °C followed by slap drying and blocking with 80 µL of Casein for 1 h at 37 °C. Afterward, plates were once again slap-dried and a 1:4 serial dilution of our immunized mouse sera was performed starting from 1:20 dilution in 30 µL of TBST and incubated at 37 °C for 1 h. Plates were then washed 4x in TBST and 30 µL of 1:5000 Goat anti-mouse IgG (H + L) Secondary Antibody HRP (ThermoFisher 62-6520) were added to each well and incubated at 37 °C for 1 h. Plates were then washed 4x in TBST and 30 uL of TMB (SeraCare) was added to each well and allowed to sit for 2 min at room temperature. TMB reactions were quenched with 30 µL of 1 N HCl and immediately read at 450 nm on a BioTek Neo2 plate reader and data plotted and fit in Prism 10 (Graphpad) using nonlinear regression sigmoidal, 4PL, X is the concentration to determine $ED_{50}$ values from curve fits. Two biological replicates each comprising two technical replicates, were carried out. Due to the shortage of sera, we were unable to conduct biological replicates for mice 2–3, 2–4, and 3–12.

For IgG subclass serological ELISAs, 30 µL of SARS-CoV-2 HexaPro S or E-69 at 3 µg/mL were placed into 384-well Nunc Maxisorp plates (ThermoFisher, 464718) in 1x TBS and incubated overnight at RT followed by slap drying and blocking with 80 µL of Casein for 1 h at 37 °C. Afterward, plates were once again slap-dried and a 1:4 serial dilution of our immunized mouse sera was performed starting from 1:80 dilution in 30 µL of TBST and incubated at 37 °C for 1 h. Plates were then washed 4x in TBST and 30 µL of 1:5000 Goat anti-mouse IgG (H + L) Secondary Antibody HRP (ThermoFisher 62-6520), Peroxidase AffiniPure™ Goat Anti-Mouse IgG Fcγ subclass 1 specific (Jackson Immuno Research, 115-035-205), Peroxidase AffiniPure™ Goat Anti-Mouse IgG Fcγ subclass 2a specific (Jackson Immuno Research, 115-035-206), and Peroxidase AffiniPure™ Goat Anti-Mouse IgG Fcγ subclass 2b specific (Jackson Immuno Research, 115-035-207) were added to each well and incubated at 37 °C for 1 h. Plates were then washed 4x in TBST and 30 uL of TMB (SeraCare) was added to each well and allowed to sit for 1 min at room temperature. TMB reactions were quenched with 30 µL of 1 N HCl and immediately read at 450 nm on a BioTek Neo2 plate reader and data plotted and fit in Prism 10 (Graphpad) using nonlinear regression sigmoidal, 4PL, X is the concentration to determine $ED_{50}$ values from curve fits. Two biological replicates each comprising two technical replicates have been carried out.

## Negative stain electron microscopy preparation, data collection, and data processing
Carbon copper formvar grids (Ted Pella 01754-F) were glow discharged using a Gloqube Plus (Quorum) at 20 mA for 30 s promptly followed by the addition of 3 µL of a $S_2$ pre-fusion constructs diluted to a concentration of 0.01 mg/mL. After 1 min the protein was aspirated using filter paper and 3 µL of 2% uranyl formate was applied and quickly removed for washing. Another 3 µL of uranyl formate was added to the grid and left to stain for 30 s before drying with filter paper and left to further air dry before imaging. Automated data collection was carried out using Leginon at a nominal magnification of 67,000 with a pixel size of 1.6 Å. Each micrograph was acquired for 500–900 ms. Negative stain data was processed using CryoSPARC. Automatic particle picking and extraction were performed using CryoSPARC for each data set. Particle images were extracted with a box size of 256 pixels with a pixel size of 1.6 Å and binned to 128 pixels for subsequent 2D classifications.

## Cryo-EM sample preparation and data collection
The E-31 cryo-EM dataset was collected over three different sessions which were combined to be processed together. 3 µL of sample was added to a glow discharged (120 s at 20 mA) UltraAuFoil R2/2:Au200 grid prior to plunge freezing using a vitrobot MarkIV (ThermoFisher Scientific) with a blot force of 0 and 6.5 sec blot time at 100% humidity and 22 °C. For E-60, 3 µL of sample was added to a glow discharged (120 s at 20 mA) UltraAuFoil R2/2:Au200 grid prior to plunge freezing using a vitrobot MarkIV (ThermoFisher Scientific) with a blot force of 0, 5.5 sec blot time, and 10 s wait time at 100% humidity and 22 °C. For E-69, 3 µL of sample was added to a glow discharged (120 s at 20 mA) UltraAuFoil R2/2:Au200 grid prior to plunge freezing using a vitrobot MarkIV (ThermoFisher Scientific) with a blot force of 0, 6 sec blot time, and 10 s wait time at 100% humidity and 22 °C.

Data were acquired using an FEI Titan Krios transmission electron microscope operated at 300 kV and equipped with a Gatan K3 direct detector and Gatan Quantum GIF energy filter, operated in zero-loss mode with a slit width of 20 eV. Automated data collection was carried out using Leginon[61] at a nominal magnification of 105,000x with a pixel size of 0.843 Å. The dose rate was adjusted to 15 counts/pixel/s, and each movie was acquired in counting mode fractionated in 75 frames of 40 ms. A total of 25,829 and 6807 micrographs were collected for E-31 and E-60 datasets, respectively. The stage was tilted 0, 30, and 45 degrees for E-31 and 20 degrees for the E-60 collection. A total of 5946 micrographs were collected for E-69 at 0 and 30 degrees tilted stage.

## Cryo-EM data processing, model building and refinement
For the E-31 structure, motion correction and contrast-transfer function (CTF) parameter estimation were performed using Warp[62] and

cryoSPARC, respectively. Automatic particle picking was performed using TOPAZ[63] and particle images were extracted with a box size of 208 pixels with a pixel size of 1.686 Å. After 2D classification and hetero-refinement using cryoSPARC[64], 1,688,203 particles were selected for cryoSPARC non-uniform refinement[65] with C3 symmetry. Particles were further subjected to another round of 2D classification (to remove particles with splayed open conformation) followed by Bayesian polishing[66] in Relion. Finally, another round of cryoSPARC non-uniform refinement with C3 symmetry and per-particle defocus refinement was carried out using the polished particles.

For the E-60 structure, motion correction, CTF estimation, automatic particle picking, and extraction were performed using Warp[62]. Particle images were extracted with a box size of 208 pixels and a pixel size of 1.686 Å. After 2D classification and hetero-refinement in cryoSPARC, 240,989 particles were selected. These particles were subjected to two rounds of 3D classification with 50 iterations each (angular sampling 7.5˚ for 25 iterations and 1.8˚ with local search for 25 iterations) using Relion[67–69]. 3D refinements were carried out using non-uniform refinement along with per-particle defocus refinement in cryoSPARC followed by Bayesian polishing[66] in Relion. Finally, another round of cryoSPARC non-uniform refinement with C3 symmetry and per-particle defocus refinement was carried out using the polished particles.

For the E-69 structure, motion correction, CTF estimation, automatic particle picking (blob picking), and extraction were performed using cryoSPARC LIVE and cryoSPARC. Particle images were extracted with a box size of 208 pixels and a pixel size of 1.686 Å. After 2D classification and hetero-refinement in cryoSPARC, 319,035 particles were selected and subjected to reference motion correction and beam tilt correction followed by a final non-uniform refinement with C3 symmetry using cryoSPARC.

Local resolution estimation, filtering, and sharpening were carried out using CryoSPARC. Reported resolutions are based on the gold-standard Fourier shell correlation (FSC) of 0.143 criterion and Fourier shell correlation curves were corrected for the effects of soft masking by high-resolution noise substitution[70,71]. UCSF Chimera[72], UCSF ChimeraX[73], and Coot[74] were used to fit and rebuild atomic models into the cryoEM maps utilizing sharpened and unsharpened maps. Model building of the rearranged region C-terminal of the fusion peptide (residues 816-861) was assisted by AlphaFold2[75,76]. The models were refined and relaxed using Rosetta[77,78] and validated using Phenix[79], Molprobity[80] and Privateer[81].

## Immunogenicity

Female BALB/c mice were purchased from Envigo (order code 047) at 7 weeks of age and were maintained in a specific pathogen-free facility within the Department of Comparative Medicine at the University of Washington, Seattle, accredited by the Association for Assessment and Accreditation of Laboratory Animal Care (AAALAC). Animal experiments were conducted in accordance with the University of Washington's Institutional Animal Care and Use Committee.

Prior to each immunization, immunogens (low endotoxin immunogen) were diluted to 20 μg/mL (2P S) or 100 μg/mL ($S_2$) in 1x PBS (1.5 mM Potassium Phosphate monobasic, 155 mM NaCl, 2.7 mM Sodium Phosphate diabasic, pH 7.4) (ThermoFisher) and mixed with 1:1 vol/vol AddaVax (InvivoGen vac-adx-10) to reach a final dose of 1 μg (2P S) or 5 μg ($S_2$) of immunogen per injection. At 8 weeks of age, 12 mice per group were injected subcutaneously in the inguinal region with 100 uL of immunogen at weeks 0, 3, 10, and 17. Group 1 received four doses of 1 μg 2P S. Group 2 received four doses of 5 μg $S_2$. Group 3 received two doses of 1 μg 2P S and boosted with two doses of 5 μg $S_2$. Mice were bled via the submental route at weeks 0, 2, 5, 12, and 19. Blood was collected in serum separator tubes (BD # 365967) and rested for 30 min at room temperature for coagulation. Serum tubes were then centrifuged for 10 min at 2000 x $g$ and serum was collected and

stored at -80 °C until use. Mouse 2–10 were euthanized before immunization 4 resulting in $n = 11$ for group2 for the reported sera ELISA, IgG isotype ELISA, and neutralization assay.

## Neutralization assays

For SARS-CoV-2 D614G S VSV, XBB.1.5 S VSV, and SARS-CoV-1 S VSV neutralization, VeroE6-TMPRSS2 cells in DMEM supplemented with 10% FBS, 1% PenStrep, and 2% Geneticin were seeded at 40,000 cells/well into 96-well plates [3610] (Corning) and incubated overnight at 37 °C. The following day, a half-area 96-well plate (Greiner) was prepared with 3-fold serial sera dilutions (starting dilutions determined for each serum and pseudovirus, 22 uL per well). An equal volume of DMEM with diluted pseudoviruses was added to each well. All pseudoviruses were diluted between 1:90–1:200 to reach a target entry of ~$10^6$ RLU. The mixture was incubated at room temperature for 45–60 min. Media was removed from the cells, and the cells were washed once with DMEM prior to the transfer of sera-pseudovirus mixture. 40 μL from each well of the half-area 96-well plate containing sera and pseudovirus were transferred to the 96-well plate seeded with cells and incubated at 37 °C for 1 h. After 1 h, an additional 40 μL of DMEM supplemented with 20% FBS and 2% PenStrep was added to the cells. After 18–20 h, 40 μL of One-Glo-EX substrate (Promega) was added to each well and incubated on a plate shaker in the dark for 5 min before reading the relative luciferase units using a BioTek Neo2 plate reader. Relative luciferase units (RLUs) were plotted and normalized in Prism (GraphPad): 100% neutralization being cells in the absence of pseudovirus and 0% neutralization being pseudovirus entry into cells without sera. Prism (GraphPad) nonlinear regression with "log[inhibitor] versus normalized response with a variable slope" was used to fit the curve. Percent neutralization was calculated by taking the interpolated percentage of entry value at a fixed dilution factor of 1/33 (v/v ratio dilution of initial sera, log(dilution factor) of 1.519) using the fit curve. 100% and 0% neutralization were defined as 0% entry and 100% entry, respectively. Calculated values from three biological replicates per sample-pseudovirus pair were used to obtain the mean percent neutralization per animal.

## Mouse challenges and virus plaque assays

7 weeks post-boost, mice (Envigo, stock# 047) were exported from the Comparative Medicine Facility at the University of Washington, Seattle, WA to an AAALAC accredited Animal Biosafety Level 3 (ABSL3) Laboratory at the University of North Carolina, Chapel Hill, where mice were acclimated for 7 days. For infection, mice were anesthetized with a mixture of ketamine/xylazine and challenged intranasally with $1 \times 10^5$ plaque-forming units (pfu) of a recombinant mouse-adapted coronavirus SARS-CoV-2 XBB.1.5 MA strain for the evaluation of vaccine efficacy (Powers et al., 2024–DOI: 10.1016/j.virusres.2024.199319) (IACUC protocol 21-272). Infected mice were monitored for daily body weight. On day 4 post-infection mice were necropsied, the degree of lung congestion was scored, and lung (caudal lobe) and nasal turbinate tissues were harvested to determine viral loads by plaque assay.

For plaque assays, the appropriate tissues were homogenized in PBS and tissue debris was pelleted at 13,000 x $g$ for 5 min. The clarified homogenates were serial-diluted and added to a confluent monolayer of Vero E6 cells (ATCC CCL-81), followed by an agarose overlay. Plaques were visualized with an overlay of Neutral Red dye on day 3 post-infection.

## Reporting summary

Further information on research design is available in the Nature Portfolio Reporting Summary linked to this article.

## Data availability

The sharpened and unsharpened cryoEM reconstructions of the prefusion-stabilized SARS-CoV-2 fusion machinery designs have been

deposited in the Electron Microscopy Data Bank (EMDB) under accession codes EMD-43435 (E-31); EMD-43436 (E-60); and EMD-43437 (E-69). The atomic models of prefusion-stabilized SARS-CoV-2 fusion machinery designs have been deposited in the Protein Data Bank (PDB) under accession codes 8VQ9 (E-31); 8VQA (E-60); and 8VQB (E-69). Other data will be available from the corresponding author upon request. For the structures that have been referenced in the text are accessible under the PDB accession code PDB 6VXX; PDB 8DYA; PDB 6XRB; PDB 5X58; and PDB 8U29. The source data underlying Figs. 1c, d, 2c, 3j–m, 4b–g, 5a–d, and Supplementary Figs. 6a, 7, 8a, b, 9a, 10a–f, and 11a, b are provided as a Source Data file. Other data will be available from the corresponding author upon request. Source data are provided in this paper.

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

## Acknowledgements

This study was supported by the National Institute of Allergy and Infectious Diseases (P01AI167966 to R.B., N.P.K and D.V., DP1AI158186 and 75N93022C00036 to D.V.), a Pew Biomedical Scholars Award (D.V.), an Investigators in the Pathogenesis of Infectious Disease Awards from the Burroughs Wellcome Fund (D.V.), a Dale F. Frey Award for Break-through Scientists from the Damon Runyon Cancer Research Foundation (T.N.S.), the University of Washington Arnold and Mabel Beckman cryoEM center and the National Institute of Health grant S10OD032290 (to D.V.). D.V. is an Investigator of the Howard Hughes Medical Institute and the Hans Neurath Endowed Chair in Biochemistry at the University of Washington.

## Author contributions

J.L. and D.V. designed the study and the experiments; J.L., C.S., and K.R.S. recombinantly expressed and purified glycoproteins. J.L. and C.S. performed binding assays. J.L. carried out pseudovirus entry assays. J.L., D.A., and Y.J.P. carried out cryoEM specimen preparation, data collection, and processing. J.L. and D.V. built and refined atomic models. E.M.L. and C.T. performed mouse immunizations and blood draws. A.S. and J.M.P. carried out a mice viral challenge and analysis. D.C. contributed unique reagents. J.L. and D.V. analyzed the data and wrote the manuscript with input from all authors; R.B., N.P.K., and D.V. supervised the project.

## Competing interests

N.P.K. and D.V. are named as inventors on patents for coronavirus nanoparticle vaccines filed by the University of Washington. N.P.K. is a paid consultant of Icosavax, Inc. and has received unrelated sponsored research agreements from Pfizer and GSK. D.C. is an employee of Vir Biotechnology and may hold shares in Vir Biotechnology. R.S.B. is a member of advisory boards for VaxArt, Takeda and Invivyd, and has collaborative projects with Gilead, J&J, and Hillevax, focused on unrelated projects.
