## [Peer Review File · Nature Communications]

A broadly generalizable stabilization strategy for sarbecovirus fusion machinery vaccinesREVIEWER COMMENTS

Reviewer #1 (Remarks to the Author):

In this work, the authors describe a protein design strategy enabling prefusion-stabilization of the SARS-CoV-2 S2 subunit and high yield recombinant expression of trimers with native structure and antigenicity, and provide proof-of-principle for fusion machinery broad spectrum sarbecovirus vaccines that motivate future development. While the study is comprehensive, I believe the manuscript can benefit from a clearer exposition of the main advancements that are being made in the work and not available from prior literature, especially in the Introduction. Overall, the work is highly relevant, and strongly merits publication and I believe, would be of significant interest to the readership of Nature Communications. I found the manuscript to be well written, and only have a few comments which the authors can address to increase the ease of readability:

Major comments:

1. While the study is comprehensive, I believe the manuscript can benefit from a clearer exposition of the main advancements that are being made in the work and not available from prior (extensive) literature, especially in the Introduction.
2. The premise of the work is that the sarbecovirus S2 subunit prefusion-stabilization strategy presented in the work represents a key step towards generating a more broadly applicable vaccine that is more resilient to viral evolution and antigenic changes due to the much higher conservation of the S2 subunit relative to the S1 subunit among SARS-CoV-2 variants and others. This, in my opinion, is best presented as a figure showing the degree of (evolutionary) conservation as it varies with the residue position, in place of pointing towards the appropriate literature references. This should further complement how the specific mutations were chosen in this work, for example for constructs such as E-31, E-69, etc., and would provide ease of readership.
3. Mutations were selected based on a prior deep mutational scanning dataset to design a prefusion S2 subunit with native quaternary structure. It is not clear to me when introducing especially multiple mutations, if epistatic effects were considered, which can significantly affect the outcome measured in terms of binding (for e.g PMID: 31591964, 37478179), stability or other aspects of the viral lifecycle including viral replicative capacity. The authors should comment on this and include appropriate citations.

Is this perhaps an effect most noticeable in the construct E-60?

Furthermore, the effect of a mutation is dependent on the sequence background due to epistasis and can vary dependent on the background leading to an evolutionary Stokes shift when subjected to selection pressure (PMID: 31591964). This can lead to significant differences in mutational effects.

4. Euler angle distributions for cryo-EM maps should be included and perhaps, 3D-FSC curves can be included in particular for maps where particle number limitations and poor rotameric configurations (such as the E-60 construct) may lead to overfitting in FSC curves and inflated resolutions.

5. The mechanism of binding (of monoclonal antibodies) can be very different between SARS-CoV-1 S2 and PRD-0038 S2 induced by the differences in the binding poses (for example, see PMID: 3747817) eliciting potentially different degrees and levels of resistance. For extrapolation to broad spectrum activity (since bound structures are not obtained), perhaps, the authors can comment on this.

Minor comments:

1. The interactions shown in Fig 1G can be better illustrated using an interaction map.

Reviewer #2 (Remarks to the Author):

This is a nicely written and high quality structure-guided study to stabilise the S2 domain of SARS-CoV-2. This will advance universal vaccine approaches for coronaviruses, as well as provide new tools for serological studies.

Only minor comments;

The authors should describe the 2P version of spike which was used to immunise mice. Only the HexaPro construct is listed in the methods section.

It would be helpful to clarify if trimerisation domains are used in the S2 constructs similar to the spike ectodomain constructs.

Fig.4b: It would be helpful to add a legend for the group icons to the right side of the figure.

It is difficult to interpret the AUC neutralisation in Fig.4e without a benchmark or positive control mAb?

Are these titers expected? Or generally low, suggesting that S2 responses are broad but non-neutralising? This context is important in helping our understanding of the contribution of non-neutralising antibody mechanisms to protection in vivo (as the authors refer to in the discussion).

Determining the isotype subclass breakdown of antibody responses would be relevant in determining if engagement with activating Fc receptors is likely with this vaccination+adjuvant platform.

Line 363-365: T cell responses following soluble recombinant protein, even with adjuvant, are generally negligible in mice. The statement should be adjusted, or references which describe T cell responses following protein-based immunisation rather than nanoparticle/viral vector/mRNA/infection should be cited.

We would like to thank the reviewers for evaluating our manuscript and for their valuable comments.

Reviewer #1 (Remarks to the Author):

In this work, the authors describe a protein design strategy enabling prefusion-stabilization of the SARS-CoV-2 S₂ subunit and high yield recombinant expression of trimers with native structure and antigenicity, and provide proof-of-principle for fusion machinery broad spectrum sarbecovirus vaccines that motivate future development. While the study is comprehensive, I believe the manuscript can benefit from a clearer exposition of the main advancements that are being made in the work and not available from prior literature, especially in the Introduction. Overall, the work is highly relevant, and strongly merits publication and I believe, would be of significant interest to the readership of Nature Communications. I found the manuscript to be well written, and only have a few comments which the authors can address to increase the ease of readability:

Thank you very much for the positive assessment of our work.

Major comments:

1. While the study is comprehensive, I believe the manuscript can benefit from a clearer exposition of the main advancements that are being made in the work and not available from prior (extensive) literature, especially in the Introduction.

Per the reviewer's suggestion, we modified the introduction to highlight prior work and current advances, as follows:

[...] Coronavirus S glycoproteins fold as spring-loaded trimers transitioning from the prefusion to the postfusion state, upon receptor engagement and proteolytic activation to promote membrane fusion and viral entry^{28–30}. This metastability, however, constitutes a challenge for producing the S₂ subunit (fusion machinery) in the prefusion state in the absence of the S₁ subunit. Prior work described prefusion-stabilizing mutations improving the biophysical properties of the S ectodomain trimer^{31–36}. Furthermore, we previously designed a fusion machinery (S₂ subunit) antigen¹⁴ stabilized through introduction of 4 out of 6 HexaPro mutations (A892P, A899P, A942P and V987P)³¹, the VFLIP inter-protomer disulfide (Y707C-T883C)³², an intra-protomer disulfide (F970C-G999C) and a C-terminus foldon trimerization domain (**Figure 1A**). The resulting construct (designated C-44) harbored protomers with a prefusion tertiary structure but a quaternary structure in which the viral membrane distal region (apex) was splayed open compared to prefusion SARS-CoV-2 S¹⁴ (**Figure 1B**). To design a prefusion S₂ subunit trimer with a native quaternary structure, we identified a set of mutations allowing high-yield recombinant production of fusion machinery trimers with native prefusion architecture and antigenicity that remained stable in various storage conditions, as revealed through structural and serological studies. [...]

2. The premise of the work is that the sarbecovirus S2 subunit prefusion-stabilization strategy presented in the work represents a key step towards generating a more broadly applicable vaccine that is more resilient to viral evolution and antigenic changes due to the much higher conservation of the S2 subunit relative to the S1 subunit among SARS-CoV-2 variants and others. This, in my opinion, is best presented as a figure showing the degree of (evolutionary) conservation as it varies with the residue position, in place of pointing towards the appropriate literature references. This should further complement how the specific mutations were chosen in this work, for example for constructs such as E-31, E-69, etc., and would provide ease of readership.

Thank you for this suggestion. We added a new panel to Figure 1A rendering sequence conservation across sarbecovirus S glycoproteins, as requested.

3. Mutations were selected based on a prior deep mutational scanning dataset to design a prefusion S2 subunit with native quarternary structure. It is not clear to me when introducing especially multiple mutations, if epistatic effects were considered, which can significantly affect the outcome measured in terms of binding (for e.g PMID: 31591964, 37478179), stability or other aspects of the viral lifecycle including viral replicative capacity. The authors should comment on this and include appropriate citations.

Is this perhaps an effect most noticeable in the construct E-60?

Furthermore, the effect of a mutation is dependent on the sequence background due to epistasis and can vary dependent on the background leading to an evolutionary Stokes shift when subjected to selection pressure (PMID: 31591964). This can lead to significant differences in mutational effects.

Epistasis is indeed a very important factor that can profoundly impact the effect of mutations depending on the 'background' in which they are introduced, as we and others described for SARS-CoV-2.

However, we would like to clarify that the designed antigens described here would not support a viral life cycle as the S₂ subunit is stabilized in the prefusion state and cannot undergo conformational changes associated with membrane fusion. Moreover, we used high-resolution structural studies to assess the accuracy with which our designed vaccine candidates recapitulated the conformation of the fusion machinery in the context of the S ectodomain trimer and retention of antigenicity with monoclonal antibodies recognizing all known antigenic sites (as shown throughout the manuscript).

This is conceptually similar to the mutations introduced in the respiratory syncytial virus vaccines recently approved for which the fusion glycoprotein is stabilized in the prefusion state and cannot refold to postfusion (PMID: 24179220).

Our data suggest that the distortions observed in the E60 construct were locally induced by the introduced mutations and resolved upon removal except for the conformation of the region downstream the fusion peptide which changes in all constructs due to the lack of S₁ subunit. We clarified the latter point in the text as follows:

“Only one region noticeably deviated from prefusion S in terms of local structure downstream of the fusion peptide (residues 833-855) (**Figure 1F and Supplementary Figure 3C**). In the intact SARS-CoV-2 S trimer, this region makes contact with the S₁ subunit C and D domains of a neighboring protomer, which are absent in our design, likely allowing adoption of an altered local conformation, as is the case in all our constructs (**Supplementary Figure 3C**).”

4. Euler angle distributions for cryo-EM maps should be included and perhaps, 3D-FSC curves can be included in particular for maps where particle number limitations and poor rotameric configurations (such as the E-60 construct) may lead to overfitting in FSC curves and inflated resolutions.

As requested, we added the angular distribution plots corresponding to each of the three cryoEM maps in Figures S2, S4 and S5. Supplementary Table 2 shows that the rotamer distribution is excellent for all three structures.

5. The mechanism of binding (of monoclonal antibodies) can be very different between SARS-CoV-1 S2 and PRD-0038 S2 induced by the differences in the binding poses (for example, see PMID: 3747817) eliciting potentially different degrees and levels of resistance. For extrapolation to broad spectrum activity (since bound structures are not obtained), perhaps, the authors can comment on this.

We previously showed that stem-helix directed antibodies and fusion peptide-directed antibodies recognize their epitopes similarly across very different viruses (PMID: 33981021, PMID: 35857703, PMID: 34344823).

We apologize but the reference provided by the reviewer points to an NMR imaging paper from 1986, which most likely resulted from a typo in the PMID provided and prevented us from commenting on possible differences of binding mechanism.

Minor comments:

1. The interactions shown in Fig 1G can be better illustrated using an interaction map.

Thank you for this suggestion. The current Fig 1G aims to show the chemical micro-environment to not only emphasize polar interactions (which can indeed also be visualized with an interaction map) but also van der Waals interactions (shape complementarity), given that most of the mutations belong to the latter category. To help the readers, we modified Fig 1 and added a semi-transparent surface representation of select side chains to better visualize shape complementarity.

Reviewer #2 (Remarks to the Author):

This is a nicely written and high quality structure-guided study to stabilise the S2 domain of SARS-CoV-2. This will advance universal vaccine approaches for coronaviruses, as well as provide new tools for serological studies.

Thank you very much for the positive assessment of our work.

Only minor comments;

The authors should describe the 2P version of spike which was used to immunise mice. Only the HexaPro construct is listed in the methods section.

As requested, we added a description of the 2P construct used to the methods section as follows:

“The SARS-CoV-2 2P S glycoprotein ectodomain construct was previously described³⁴ and comprises an abrogated S₁/S₂ cleavage site (R682S, R683G and R685G)^{28,30,55,56}, two consecutive proline stabilizing mutations (K986P and V987P, so called 2P^{57,58}) and a C-terminal foldon trimerization domain⁵⁹.”

It would be helpful to clarify if trimerisation domains are used in the S2 constructs similar to the spike ectodomain constructs.

We apologize for any confusion and have now clarified that all constructs used a C-terminal foldon fusion in the main text and in the Methods section.

Fig.4b: It would be helpful to add a legend for the group icons to the right side of the figure.

We added the requested legend to Figure 4B-G.

It is difficult to interpret the AUC neutralisation in Fig.4e without a benchmark or positive control mAb? Are these titers expected? Or generally low, suggesting that S2 responses are broad but non-neutralising? This context is important in helping our understanding of the contribution of non-neutralising antibody mechanisms to protection in vivo (as the authors refer to in the discussion).

We apologize for any confusion. To improve clarity, we re-rendered (i) Figure 4 B-D (binding) by adding bar graphs representing the mean and standard deviation of each group and (ii) Figure 4 E-G (neutralization) by presenting the fraction of neutralizing activity achieved at a 1/33 serum dilution for easier comparisons.

Group 1, for which animals were only immunized with prefusion-stabilized SARS-CoV-2 2P S, is our control and benchmark for this study. As explained in the text, neutralizing activity elicited by our lead prefusion S₂ antigen is more modest than that of prefusion S but much less affected by mutations found in SARS-CoV-2 variants.

This is explained in the section titled A prefusion-stabilized SARS-CoV-2 fusion machinery vaccine elicits broadly reactive antibody responses:

“SARS-CoV-2 2P S or SARS-CoV-2 2P S followed by E-69 vaccination elicited potent neutralizing antibodies against the vaccine-matched SARS-CoV-2 Wu/G614 whereas E-69 elicited modest neutralizing activity. SARS-CoV-1 S VSV neutralization was highest upon vaccination with SARS-CoV-2 2P S and comparably lower for the groups vaccinated with E-69 or SARS-CoV-2 2P S followed by E-69. Neutralization of XBB.1.5 VSV S, however, was marginally higher for mice vaccinated with E-69 or with SARS-CoV-2 2P S followed by E-69 as compared to vaccination with SARS-CoV-2 2P S.”

And in the Discussion:

“Given the limited potency of known fusion machinery-directed monoclonal antibodies^{18,19,21,22,24}, relative to S₁-targeting antibodies, the plasma neutralizing activity induced by SARS-CoV-2 S₂ was weaker than that of prefusion S 2P although its breadth was much less affected by antigenic changes. In vivo evaluation of vaccine efficacy suggests that SARS-CoV-2 S₂ protected mice comparably to prefusion S 2P, based on weight loss and viral replication in the upper and lower airways, upon challenge with the immune evasive SARS-CoV-2 XBB.1.5 variant.”

Determining the isotype subclass breakdown of antibody responses would be relevant in determining if engagement with activating Fc receptors is likely with this vaccination+adjuvant platform.

We added ELISA data that was conducted against E-69 to determine IgG subclasses in the serum of mice in the three vaccine groups. The analysis can be found in Supplementary Figure 10 and is described in the revised manuscript as follows:

“ELISA analysis of vaccine-elicited serum antibodies five weeks post dose 4 (week 22) targeting SARS-CoV-2 E-69 showed that immunization with E-69 or with E-69 followed by SARS-CoV-2 2P S (GMTs: 4.9/4.8) led to slightly higher IgG binding titers than SARS-CoV-2 2P S immunization (GMT: 4.4), providing proof-of-principle for specific elicitation of fusion machinery-directed antibodies (**Supplementary Figure 10**). Moreover, E-69 vaccination elicited a broad spectrum of IgG subclasses, including IgG1, IgG2a and IgG2b, that was comparably balanced to that induced by SARS-CoV-2 2P S vaccination (**Supplementary Figure 10**).”

Supplementary Figure 10. Analysis of vaccine-elicited serum binding titers for different IgG subclasses against SARS-CoV-2 E-69. a-d, Analysis of IgG (a), IgG1 (b), IgG2a (c), and IgG2b (d) binding titers against E-69 analyzed by ELISAs using sera obtained five weeks post dose 4. Geometric means are shown as bars with SD. Each data point represents the mean of two biological replicates each comprising two technical replicates. e, QQ plots of residuals for the mean of log(ED50) values from two biological replicates of ELISA. f, Dose-response curves of serum antibody binding to E-69. Each data point represents the mean of two technical replicates and SD are shown with bars. One representative out of two biological replicates is shown. Comparisons between multiple groups for (a-d) were made by ordinary one-way analysis of variance (ANOVA) followed by Tukey's multiple comparisons test. ns : $P > 0.05$, * : $P \leq 0.05$, ** : $P \leq 0.01$, *** : $P \leq 0.001$, **** : $P \leq 0.0001$.

Line 363-365: T cell responses following soluble recombinant protein, even with adjuvant, are generally negligible in mice. The statement should be adjusted, or references which describe T cell responses following protein-based immunisation rather than nanoparticle/viral vector/mRNA/infection should be cited.

We completely agree that protein subunit vaccines induce very weak (if any) CD8 T cell responses. We removed the statement about T cells on line 363-365 from the discussion section as our data suggest that Fc-mediated effector functions most likely played a role here.

REVIEWERS' COMMENTS

Reviewer #1 (Remarks to the Author):

In this revised manuscript, the authors have satisfactorily addressed all of the reviewers' suggestions/comments. I would strongly support the manuscript for publication.

Reviewer #2 (Remarks to the Author):

I am satisfied that the authors have addressed all of my comments.